# End to End Trainable Active Contours via Differentiable Rendering

**Shir Gur & Tal Shaharabany**
School of Computer Science, Tel Aviv University

**Lior Wolf**
Facebook AI Research and Tel Aviv University

## Abstract

We present an image segmentation method that iteratively evolves a polygon. At each iteration, the vertices of the polygon are displaced based on the local value of a 2D shift map that is inferred from the input image via an encoder-decoder architecture. The main training loss that is used is the difference between the polygon shape and the ground truth segmentation mask. The network employs a neural renderer to create the polygon from its vertices, making the process fully differentiable. We demonstrate that our method outperforms the state of the art segmentation networks and deep active contour solutions in a variety of benchmarks, including medical imaging and aerial images. Our code is available at https://github.com/shirgur/ACDRNet.

## 1 Introduction

The importance of automatic segmentation methods is growing rapidly in a variety of fields, such as medicine, autonomous driving and satellite image analysis, to name but a few. In addition, with the advent of deep semantic segmentation networks, there is a growing interest in the segmentation of common objects with applications in augmented reality and seamless video editing.

Since the current semantic segmentation methods often capture the objects well, except for occasional inaccuracies along some of the boundaries, fitting a curve to the image boundaries seems to be an intuitive solution. Active contours, is a set of techniques that given an initial contour (which can be provided by an existing semantic segmentation solution) grow iteratively to fit an image boundary. Active contour may also be appropriate in cases, such as medical imaging, where the training dataset is too limited to support the usage of a high-capacity segmentation network.

Despite their potential, the classical active contours fall behind the latest semantic segmentation solutions with respect to accuracy. The recent learning-based active contour approaches were not yet demonstrated to outperform semantic segmentation methods across both medical datasets and real world images, despite having success in specific settings.

In this work, we propose to evolve an active contour based on a 2-channel displacement field (corresponding to 2D image coordinates) that is inferred directly and only once from the input image. This is, perhaps, the simplest approach, since unlike the active contour solutions in literature, it does not involve and balance multiple forces, and the displacement is given explicitly. Moreover, the architecture of the method is that of a straightforward encoder-decoder with two decoding networks. The loss is also direct, and involves the comparison of two mostly binary images.

The tool that facilitates this explicit and direct approach is a neural mesh renderer. It allows for the propagation of the intuitive loss, back to the displacement of the polygon vertices. While such renderers have been discovered multiple times in the past, and were demonstrated to be powerful solutions in multiple reconstruction problems, this is the first time, as far as we can ascertain that this tool is used for image segmentation.

Our empirical results demonstrate state of the art performance in a wide variety of benchmarks, showing a clear advantage over classical active contour methods, deep active contour methods, and modern semantic segmentation methods.

## 2 RELATED WORK

**Neural Renderers** A neural mesh renderer is a fully differential mapping from a mesh to an image. While rendering the 3D or 2D shape given vertices, faces, and face colors is straightforward, the process involves sampling on a grid, which is non-differentiable. One obtains differentiable rendering, by sampling in a smooth (blurred) manner (Kato et al., 2018) or by approximating the gradient based on image derivatives, as in (Loper & Black, 2014). Such renderers allow one to Backpropagate the error from the obtained image back to the vertices of the mesh.

In this work we employ the mesh renderer of Kato et al. (2018). Perhaps the earliest mesh renderer was presented by Smelyansky et al. (2002). Recent non-mesh differential renders include the point cloud renderer of Insafutdinov & Dosovitskiy (2018) and the view-based renderer of Eslami et al. (2018). Gkioxari et al. (2019) use a differentiable sampler to turn a 3D mesh to a point cloud and solve the task of simultaneously segmenting a 2D image object, while performing a 3D reconstruction of that object. This is a different image segmentation task, which unlike our setting requires a training set of 3D models and matching 2D views.

**Active contours** Snakes were first introduced by Kass et al. (1988), and were applied in a variety of fields, such as lane tracking (Wang et al., 2004), medicine (Yushkevich et al., 2006) and image segmentation (Michailovich et al., 2007). Active contours evolve by minimizing an energy function and moving the contour across the energy surface until it halts. Properties, such as curvature and area of the contour, guide the shape of the contour, and terms based on the image gradients or various alternatives attract contours to edges. Most active contour methods rely on an initial contour, which often requires a user intervention.

Variants of this method have been proposed, such as using a balloon force to encourage the contour to expand and help with bad initialization, *e.g.* contours located far from objects (Kichenassamy et al., 1995; Cohen, 1991). Kichenassamy et al. (1995) employ gradient flows, modifying the shrinking term by a function tailored to the specific attracting features. Caselles et al. (1997) presented the Geodesic Active Contour (GAC), where contours deform according to an intrinsic geometric measure of the image, induced by image features, such as borders. Other methods have replaced the use of edge attraction by the minimization of the energy functional, which can be seen as a minimal partition problem (Chan & Vese, 2001; Marquez-Neila et al., 2014).

The use of learning base models coupled with active contours was presented by Rupprecht et al. (2016), who learn to predict the displacement vector of each point on the evolving contour, which would bring it towards the closest point on the boundary of the object of interest. This learned network relies on a local patch that is extracted per each contour vertex. This patch moves with the vertex as the contour evolves gradually. Our method, in contrast, predicts the displacement field for all image locations at once. This displacement field is static and it is the contour which evolves. In our method, learning is not based on a supervision in the form of the displacement to the nearest ground truth contour but on the difference of the obtained polygon from the ground truth shape.

A level-set approach for active contours has also gained popularity in the deep learning field. Work such as (Wang et al., 2019; Hu et al., 2017; Kim et al., 2019) use the energy function as part of the loss in supervised models. Though fully differentiable, the level-sets do not enjoy the simplicity of polygons and their ability to fit straight structures and corners that frequently appear in man-made objects (as well as in many natural structures). The use of polygon base snakes in neural networks, as a fully differentiable module, was presented by Marcos et al. (2018); Cheng et al. (2019) in the context of building segmentation.

**Building segmentation and recent active contour solutions** Semi-automatic methods for urban structure segmentation, using polygon fitting, have been proposed by Wang et al. (2006); Sun et al. (2014). Wang et al. (2016); Kaiser et al. (2017) closed the gap for full automation, overcoming the unique challenges in building segmentation. Marcos et al. (2018) argued that the geometric properties, which define buildings and urban structures, are not preserved in conventional deep learning semantic segmentation methods. Specifically, sharp corners and straight walls, are only loosely

learned. In addition, a pixel-wise model does not capture the typical closed-polygon structure of the buildings. Considering these limitations, Marcos et al. (2018) presented the Deep Structured Active Contours (DSAC) approach, in order to learn a polygon representation instead of the segmentation mask. The polygon representation of Active Contour Models (ACMs) is well-suited for building boundaries, which are typically relatively simple polygons. ACMs are usually modeled to attract to edges of the reference map, mainly the raw image, and penalize over curvature regions. The DSAC method learns the energy surface, to which the active contour attracts. During training, DSAC integrates gradient propagation from the ACM, through a dedicated structured-prediction loss that minimizes the Intersection-Over-Union.

Cheng et al. (2019), extended the DSAC approach, presenting the Deep Active Ray Network (DARNet), based on a polar representation of active contours, also known as active rays models of Denzler & Niemann (1999). To handle the complexity of rays geometry, Cheng et al. (2019) reparametrize the rays and compute L1 distance between the ground truth polygon, represented as rays and the predicted rays. Furthermore, in order to handle the non-convex shapes inherit in the structures, Cheng et al. (2019) presented the Multiple Sets of Active Rays framework, which is based on the deep watershed transform by Bai & Urtasun (2017).

Concurrently to our work, Hatamizadeh et al. (2019) presented DCAC, a Deep Convolutional Active Contours method which extends DSAC, and performs instance segmentation in aerial images using a differential renderer.

Castrejón et al. (2017) and Acuna et al. (2018) presented a polygonal approach for objects segmentation. Given an image patch, the methods use a recurrent neural network to predict the ground truth polygon. Acuna et al. (2018) additionally made use of graph convolution networks and a reinforcement learning approach. Ling et al. (2019), introduced the Curve-GCN, an interactive or fully-automated method for polygon segmentation, learning an embedding space for points, and using Graph Convolution neural networks to estimate the amount of displacement for each point. Ling et al. (2019) also presented the use of differentiable rendering process, using the non-differentiable OpenGL renderer, and the method of Loper & Black (2014) for computing the first order Tylor approximation of the derivatives. Curve-GCN supports both polygon and spline learning, but also added supervision by learning an explicit location for each point, and edge maps.

Both DSAC and DARNet enjoy the benefit of a polygon-based representation. However, this is done using elaborate and sophisticated schemes. Curve-GCN on the other hand, benefits from the use of differentiable rendering, but suffers from a time consuming mechanism that restricts it to apply it only in the fine-tuning stage. Our method is considerably simpler due to the use of a fast differentiable renderer, and a 2-D displacement field in the scale of the input. It uses only ground truth masks as supervision, and two additional loss terms that are based on time-tested pulling forces from the classical active contour literature: the Balloon and Curvature terms.

## 3 METHOD

Our trained network includes an encoder $E$ and decoder $D$. In addition, a differential renderer $R$ is used, as well as a triangulation procedure $L$. In this work we use the Delaunay triangulation.

Let $S$ be the set of all training images. Given an image $I \in \mathbb{R}^{c \times h \times w}$, an initial polygon contour $P^0$ is produced by an oracle $A$, and the faces of this shape are retrieved from a triangulation procedure $L$, which returns a list $F$ of mesh faces, each a triplet of polygon vertex indices. In many benchmarks in which the task is to segment given a Region of Interest (ROI) extracted by a detector or marked by a user, the oracle simply returns a fixed circle in the middle of the ROI. Fig. 3 illustrates the initial contour generation process.

The contour evolves for $T$ iterations, from $P^0$ to $P^1$ and so on until the final shape given by $P^T$. It is represented by a list of $k$ vertices $P^t = [p_1^t, p_2^t, \ldots, p_k^t]$, where each vertex is a two dimensional coordinate. This evolution follows a dual-channel displacement field:

$$J = D_1(E(I)) \in \mathbb{R}^{c \times h \times w}. \tag{1}$$

For every vertex $j = 1..k$, the update follows the following rule:

$$p_j^t = p_j^{t-1} + J(p_j^{t-1}) \tag{2}$$

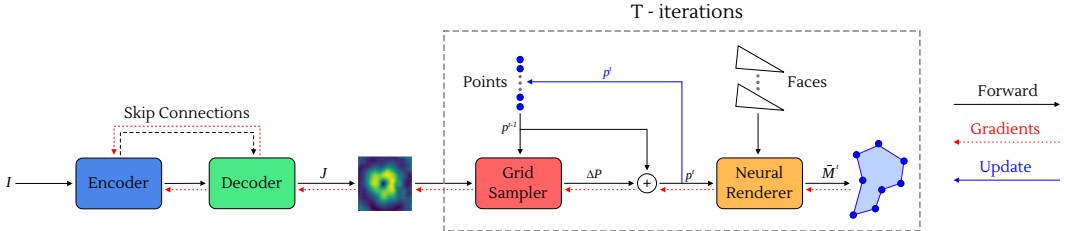

Figure 1: Illustration of our method. The input image $I$ is encoded using network $E$ and decoded back by the decoder $D$ to provide a 2D displacement field $J$. The vertices of the polygon at time $t-1$ are updated by the displacement values specified by $J$, creating the polygon of the next time step. During training, a neural renderer reconstructs the polygon shape, based on the polygon vertices and the output of triangulation process. A loss is provided by comparing the reconstructed shape with the ground truth segmentation mask.

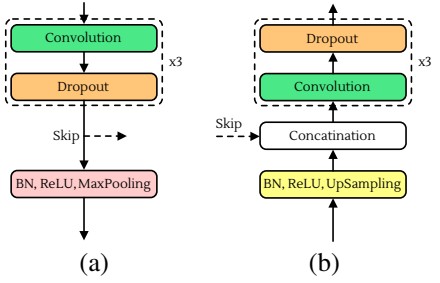

Figure 2: Illustration of the (a) encoder and (b) decoder blocks.

Figure 3: Illustration of the initial contour generation process.

where $J(p_j^{t-1})$ is a 2D vector, and the operation $J(\cdot)$ denotes the sampling operation of displacement field $J$, using bi-linear interpolation at the coordinates of $p_j^{t-1}$.

Coordinates which fall outside the boundaries of the image are then truncated (the following uses square brackets to refer to indexed vector elements):

$$p_j^t[1] = min(h, max(0, p_j^t[1])), p_j^t[2] = min(w, max(0, p_j^t[2])) \tag{3}$$

The neural renderer, given the vertices and the faces returns the polygon shape as a mask, where all pixels inside the faces $F$ are ones, and zero otherwise:

$$\bar{M}^t = R(P^t, F) \in \mathbb{R}^{h \times w} \tag{4}$$

where $\bar{M}^t$ is the output segmentation at iteration $t$. This mask is mostly binary, except for the boundaries where interpolation occurs. Because the used renderer works in 3D space, we project all points to 3D by setting their $z$ axis to 1, and use orthogonal projection.

This segmentation mask $\bar{M}^t$ is compared to the ground truth mask $M$ at each iteration, and accumulated over $T$ iterations to obtain the segmentation loss:

$$\mathcal{L}_{\text{SEG}} = \sum_{t=1}^{T} \|\bar{M}^t - M\|_2 \tag{5}$$

where $\| \cdot \|_2$ is the MSE loss applied to all mask values.

The curve evolution in classic active contour models is influenced by two additional forces: a ballooning force and a curvature minimizing force. In our feed forward network, these are manifested as training losses. The Balloon term $\mathcal{L}_\mathcal{B}$, maximizes the polygon area, causing it to expand:

$$\mathcal{L}_\mathcal{B} = \frac{1}{h \times w} \sum_{x} (1 - \bar{M}^t(x)) \tag{6}$$

**Algorithm 1** Active contour training of networks $E, D$. Shown for a batch size of one.

---

**Require:** $\{I_i\}_{i=1}^n$ : Input images, $\{M_i\}$ : Matching ground truth segmentation masks, $A$ : Initial guess oracle, $R$ : differential renderer, $L$ : a triangulation procedure, $k$ : number of vertices, $T$ : number of iterations, $\lambda_1, \lambda_2$ : a weighting parameter.

1:  Initialize networks $E, D$
2:  **for** multiple epochs **do**
3:     **for** i = 1....n **do**
4:        $P^0 = [p_1^0, p_2^0, .., p_k^0] \leftarrow A(I_i)$              ▷ Initialize the polygon using the oracle
5:        $F \leftarrow L(P^0)$                       ▷ Triangulation to obtain the mesh faces
6:        $J \leftarrow D(E(I))$
7:        **for** t = 1....T **do**
8:           Let $P^t = [p_1^t, p_2^t, .., p_k^t]$
9:           **for** j = 1....k **do**
10:             $p_j^t \leftarrow p_j^{t-1} + J(p_j^{t-1})$          ▷ Set the vertices of polygon $P^t$
11:          $\bar{M}^t = R(P^t, F)$              ▷ The polygon shape as an image
12:          $\mathcal{L} \leftarrow \mathcal{L} + \|\bar{M}^t - M\|_2 + \lambda_1 \frac{1}{h \times w} \sum_x (1 - \bar{M}^t(x)) + \lambda_2 \frac{1}{k} \sum_j |p_{j-1}^t - 2p_j^t + p_{j+1}^t|_2$

13:        Backpropagate the loss $\mathcal{L}$ and update $E, D$

---

where $h$ and $w$ are the segmentation height and width, and $x$ denotes a single pixel in $\bar{M}^t$. Second, the Curvature term $\mathcal{L}_\mathcal{K}$ minimizes the curvature of the polygon, resulting in a more smooth form:

$$\mathcal{L}_\mathcal{K} = \frac{1}{k} \sum_j ||p_{j-1}^t - 2p_j^t + p_{j+1}^t||_2 \tag{7}$$

where the $L_2$ norm is computed on 2D coordinate vectors.

The complete training loss is therefore:

$$\mathcal{L} = \mathcal{L}_{\text{SEG}} + \lambda_1 \mathcal{L}_\mathcal{B} + \lambda_2 \mathcal{L}_\mathcal{K}, \tag{8}$$

for some weighting parameter $\lambda_1$ and $\lambda_2$. It is applied after each evolution of the contour (and not just on the final contour). See Alg. 1 for a listing of the process.

### 3.1   Architecture and training

We employ an Encoder-Decoder architecture with U-Net skip connections (Çiçek et al., 2016), which link layers of matching sizes between the encoder sub-network and the decoder sub-network, as can be seen in Fig. 1. The encoder part, as can be seen in Fig. 2(a), is built from blocks which are mirror-like versions of the relative decoder blocks, as can be seen in Fig. 2(b), connected by a skip connection.

An encoder block consists of (i) three sub-blocks of convolution layer followed by dropout with probability of 0.2, (ii) ReLU, (iii) batch normalization and (iv) max-pooling, to down-sample the input feature map.

A decoder block consists of (i) batch normalization, (ii) ReLU, (iii) bi-linear interpolation, which up-samples the input feature map to the size of the skip connection, (iv) concatenation of the input skip connection and the output of the previous step, (v) three sub-blocks of convolution layer, followed by dropout with probability of 0.2. For the last decoder block, we omit the dropout layer, and up-sample to the input image size using bi-linear interpolation. To get the pixel-wise probabilities, we employ the Sigmoid (logistic) activation.

For the initial contour, unlike DARNet (Cheng et al., 2019) which multiple initializations, we simply use a fixed circle centered at the middle of the input image, with a diameter of 16 pixels, across all datasets.

For training the segmentation networks, we use the ADAM optimizer Kingma & Ba (2014) with a learning rate 0.001, batch size varies depending of input image size, for $64 \times 64$ we use 100, $128 \times 128$ we use 50. We set $\lambda_1 = 10^{-2}$ and $\lambda_2 = 5 \cdot 10^{-1}$.

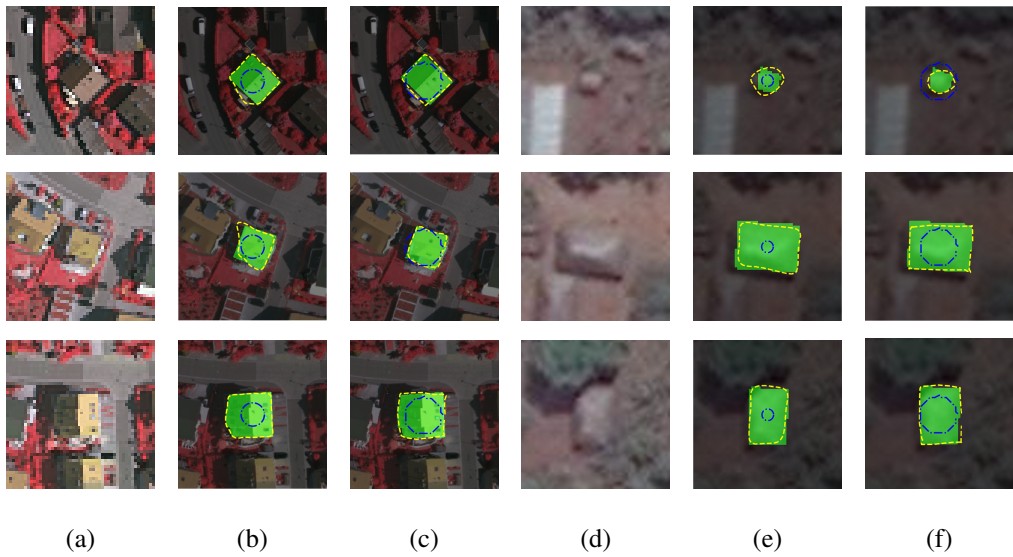

|       |       |       |       |       |       |
|  (a)  |  (b)  |  (c)  |  (d)  |  (e)  |  (f)  |

Figure 4: Qualitative results of DARNet (Cheng et al., 2019) and our method. Columns **(a)-(c)** show results from the Vaihingen dataset (Rottensteiner et al.), and **(d)-(f)** show results from the Bing huts (Marcos et al., 2018) dataset. **(b) and (e)** - DARNet (Cheng et al., 2019), **(c) and (f)** - Ours. **Blue** - Initial contour. **Yellow** - Final contour. **Green** - GT Mask.

Table 1: Quantitative results on the two buldings datasets of Vaihingen (Rottensteiner et al.) and Bing (Marcos et al., 2018). † denotes the use of DSAC as backbone, and ‡ denotes the use of DARNet as backbone.

| Method | Vaihingen | | | | Bing | | | |
| | F1-Score | mIoU | WCov | BoundF | F1-Score | mIoU | WCov | BoundF |
|---|---|---|---|---|---|---|---|---|
| FCN† | - | 81.09 | 81.48 | 64.6 | - | 69.88 | 73.36 | 30.39 |
| FCN‡ | - | 87.27 | 86.89 | 76.84 | - | 74.54 | 77.55 | 37.77 |
| DSAC† | - | 71.10 | 70.76 | 36.44 | - | 38.74 | 44.61 | 37.16 |
| DSAC‡ | - | 60.37 | 61.12 | 24.34 | - | 57.23 | 63.09 | 15.98 |
| DARNet‡ | 93.65 | 88.24 | 88.16 | 75.91 | 85.21 | 75.29 | 77.07 | 38.08 |
| Ours | **95.62** | **91.74** | **89.03** | **79.19** | **91.04** | **84.73** | **82.23** | **58.29** |

## 4 EXPERIMENTS

For evaluation, we use the common segmentation metrics of F1-score and Intersection-over-Union (IoU). Additionally, for the buildings segmentation datasets, we use the Weighted Coverage (WCov) and Boundary F-score (BoundF), which is the averaged F1-score over thresholds from 1 to 5 pixels around the ground truth, as described by Cheng et al. (2019).

### 4.1 BUILDING SEGMENTATION

We consider two publicly available datasets in order to evaluate our method, the Vaihingen (Rottensteiner et al.) dataset, which contains buildings from a German city, and the Bing Huts dataset (Marcos et al., 2018), which contains huts from a village in Tanzania. A third dataset named TorontoCity, proposed by Marcos et al. (2018); Cheng et al. (2019) is not yet publicly available. **The Vaihingen dataset** consists of 168 buildings extracted from ISPRS Rottensteiner et al.. All images contain centered buildings with a very dense environment, including other structures, streets, trees and cars, which makes the task more challenging. The dataset is divided into 100 buildings for training, and the remaining 68 for testing. The image's size is $512 \times 512 \times 3$, which is relatively high. We experiment with different resizing factors during training. **The Bing Huts dataset** consists of 606 images,

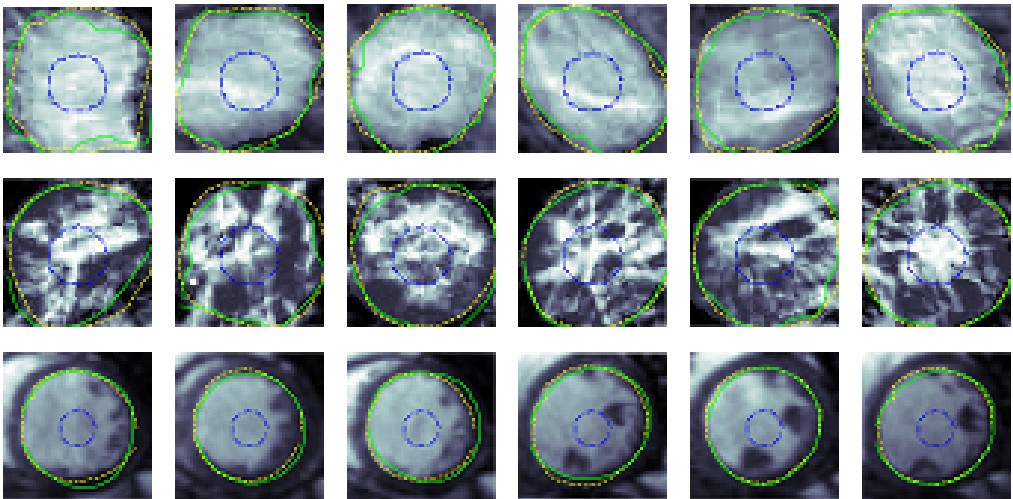

Figure 5: Qualitative results on the mammographic and cardiac datasets. **Top** - INBreast (Moreira et al., 2012), **Middle** - DDSM-BCRP (Heath et al., 1998), **Bottom** - SCD (Radau et al., 2009). **Blue** - Initial contour. **Yellow** - Final contour. **Green** - GT Mask.

Table 2: Quantitative results on the two mammographic datasets of INBreast (Moreira et al., 2012) and DDSM-BCRP (Heath et al., 1998). Reported results are the F1-Score.

| Method | INBreast | DDSM-BCRP |
|---|---|---|
| Ball & Bruce (2007) | 90.90 | 90.00 |
| Zhu et al. (2018) | 90.97 | 91.30 |
| Li et al. (2018) | 93.66 | 91.14 |
| Singh et al. (2020) | 92.11 | - |
| Ours | **94.28** | **92.32** |

Table 3: Results on the cardiac MR Left Ventricle segmentation dataset of SCD (Radau et al., 2009), F1-Score on the entire test set.

| Method | F1-Score |
|---|---|
| Queirós et al. (2014) | 0.90 |
| Liu et al. (2016) | 0.92 |
| Avendi et al. (2016) | 0.94 |
| Ngo et al. (2017) | 0.88 |
| Ours | **0.95** |

335 images for train and 271 images for test. The images suffer from low spatial resolution and have the size of $64 \times 64$, in contrast to the Vaihingen dataset.

We compare our method to the relevant previous works, following the evaluation process, as described in Cheng et al. (2019), using the published test/val/train splits. The evaluated polygons are scaled, according to the original code of Cheng et al. (2019). For both datasets, we augment the training data (of the networks) by re-scaling in factors of $[0.75, 1, 1.25, 1.5]$, and rotating by $[0, 15, 45, 60, 90, 135, 180, 210, 240, 270]$ degrees.

As can be seen in Tab. 1 our method significantly outperforms the baseline methods on both building datasets. Fig. 4 compares the results of our method with the leading method by Cheng et al. (2019).

## 4.2 MEDICAL IMAGING

We evaluate our method using two common mammographic mass segmentation datasets, the IN-Breast (Moreira et al., 2012), DDSM-BCRP (Heath et al., 1998), and a cardiac MR left ventricle segmentation datasets, the SCD (Radau et al., 2009). For the mammographic dataset, we follow previous work and use the expert ROIs, which were manually extracted, and the same train/test split as Zhu et al. (2018); Li et al. (2018). **INBreast dataset** consists of 116 accurately annotated masses, with mass size ranging from $15mm^2$ to $3689mm^2$. The dataset is divided into into 58 images for train and 58 images for test, as done in previous work. **DDSM-BCRP dataset** consists of 174 annotated masses, provided by radiologists. The dataset is divided into into 78 images for train and 5788 images for test, as done in previous work. **SCD dataset** The Sunnybrook Cardiac Data (SCD),

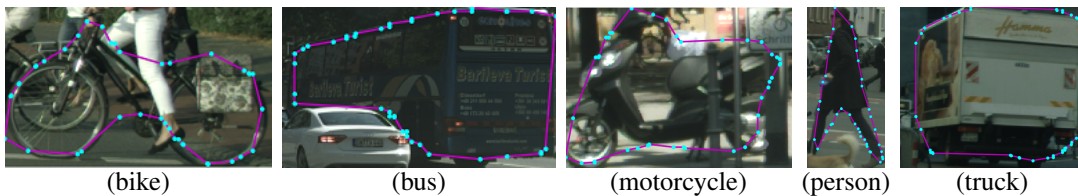

| (bike) | (bus) | (motorcycle) | (person) | (truck) |

Figure 6: Sample results from the Cityscapes dataset.

| Model | Bicycle | Bus | Person | Train | Truck | Motorcycle | Car | Rider | Mean |
|---|---|---|---|---|---|---|---|---|---|
| Polygon-RNN++ (with BS) | 63.06 | 81.38 | 72.41 | 64.28 | 78.90 | 62.01 | 79.08 | 69.95 | 71.38 |
| PSP-DeepLab | 67.18 | 83.81 | 72.62 | 68.76 | **80.48** | 65.94 | 80.45 | 70.00 | 73.66 |
| Polygon-GCN (with PS) | 66.55 | 85.01 | 72.94 | 60.99 | 79.78 | 63.87 | 81.09 | 71.00 | 72.66 |
| Spline-GCN (with PS) | 67.36 | **85.43** | 73.72 | 64.40 | 80.22 | 64.86 | 81.88 | 71.73 | 73.70 |
| Ours | **68.08** | 83.02 | **75.04** | **74.53** | 79.55 | **66.53** | **81.92** | **72.03** | **75.09** |

Table 4: **Cityscapes dataset.** Quantitative results reported in mean IoU. **BS** indicates that the model uses beam search, **PS** indicates that the model train with explicit points supervision.

the MICCAI 2009 Cardiac MR Left Ventricle Segmentation Challenge data, consist of 45 cine-MRI images from a mix of patients and pathologies. The dataset is split into three groups of 15, resulting in about 260 2D-images each, and report results for the endocardial segmentation.

Tab. 2 and 3 show that our method outperforms all baseline methods on the three medical imaging benchmarks. Sample results for our method are shown in Fig. 5.

## 4.3 STREET VIEW IMAGES

Following Ling et al. (2019), we employ the Cityscapes dataset (Cordts et al., 2016) to evaluate our model in the task of segmenting street images. The dataset consists of 5000 images, and the experiments employ the train/val/test split of Castrejon et al. (2017). Results are evaluated using the mean IoU metric.

The Cityscapes dataset, unlike the other datasets, contains single objects with multiple contours. Similarly to Curve-GCN (Ling et al., 2019), we first train our model on to segment single contours, given the patch that contains that contour. Then, after the validation error stops decreasing, we train the final network to segment all contours from the patch that contains the entire object. Note that the network outputs a single output contour for the entire instance. We follow the same augmentation process as presented for the buildings datasets, using an input resolution of $64 \times 64$.

We compare our method with previous work on semantic segmentation, including PSP-DeepLab (Chen et al., 2017), and the polygon segmentation methods Polygon-RNN++ (Acuna et al., 2018) and Curve-GCN (Ling et al., 2019).

As can be seen in from Tab. 4, our model outperforms and all other methods in six out of eight categories, and achieves the highest average performance across classes by a sizable margin that is bigger than the differences between the previous contributions (the previous methods achieve average IoU of 71.38–73.70, we achieve 75.09) . Unlike Curve-Net (Ling et al., 2019), we do not use additional supervision in the form of explicit point location and edge maps. Sample results can be seen in Fig. 6.

## 4.4 MODEL SENSITIVITY

To evaluate the sensitivity of our method to the key parameters, we varied the number of nodes in the polygon and the number of iterations. Both parameters are known to effect active contour models.

**Number of Vertices** We experimented with different number of vertices, from simple to complex polygons - [4, 8, 16, 32, 64, 128]. In Fig. 8 - top row, we report the Dice and mIoU on all datasets,

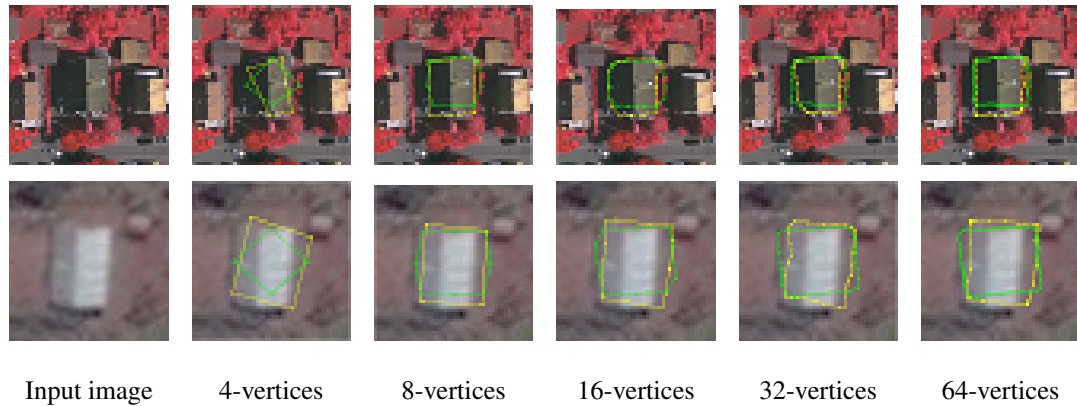

| Input image | 4-vertices | 8-vertices | 16-vertices | 32-vertices | 64-vertices |

Figure 7: Varying number of vertices. **Yellow** - Our method. **Green** - DARNet (Cheng et al., 2019)

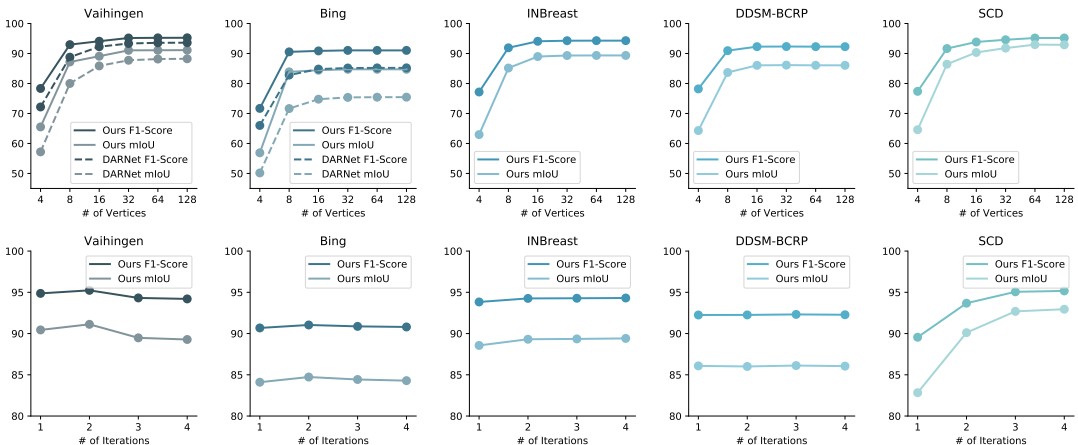

Figure 8: **Top** - different number of vertices, and **Bottom** - different number of iterations. Results for DARNet (Cheng et al., 2019) are available for the buildings datasets.

including results on DARNet (Cheng et al., 2019) on their evaluation datasets. As can be seen, segmenting with simple polygons yields lower performance, while as the number of vertices increases the performance quickly saturated at about 32 vertices. A clear gap in performance is visible between our method and DARNet (Cheng et al., 2019), especially with low number of vertices. Fig. 7 illustrates that gap on the two buildings datasets.

**Number of Iterations** As can be seen from Fig. 8(Bottom), the effect of the iterations number is show to be moderate, although a mean increase is seen over all datasets, saturating at about 3 iterations. It is also visible that our model, trained only by a single iteration, has learned to produce a displacement map that manipulate the initial circle close to optimal.

**Training Image Resolution** All of our models are trained at the resolution of $64 \times 64$ pixels. We have experimented with different input resolutions on the Vaihingen dataset. As can be seen in Tab. 5, there is a steep degradation in performance below $64$, which we attribute to the lack of details. When doubling the resolution to $128 \times 128$, the performance slightly degrades, however, that model could improve with further training epochs.

**Ablation Study** In Tab. 6 we show the effect of different loss combinations on our model performance on the Vaihingen benchmark. The compound loss $\mathcal{L}$ is better than its derivatives. Each the ballooning loss improves performance over not using auxiliary losses at all, while the curvature loss by itself does not. We note that even without the auxiliary losses, with a single straightforward loss term, our method outperforms the state of the art.

Table 5: Evaluation of different image resolutions during training on the Vaihingen dataset.

| Resolution | 16 | 32 | 64 | 128 |
|---|---|---|---|---|
| F1-Score | 13.11 | 54.86 | **95.62** | 95.13 |
| mIoU | 7.13 | 39.50 | **91.74** | 90.85 |

Table 6: Evaluation of different loss combinations on the Vaihingen dataset.

| Loss combination | $\mathcal{L}_{\text{SEG}}$ | $\mathcal{L}_{\text{SEG}} + \mathcal{L}_{\mathcal{K}}$ | $\mathcal{L}_{\text{SEG}} + \mathcal{L}_{\mathcal{B}}$ | $\mathcal{L}$ |
|---|---|---|---|---|
| F1-Score | 94.94 | 94.80 | 95.13 | **95.62** |
| mIoU | 90.31 | 90.20 | 90.82 | **91.74** |

## 5 CONCLUSIONS

Active contour methods that are based on a global neural network inference hold the promise of improving semantic segmentation by means of an accurate edge placement. We present a novel method, which could be the most straightforward active contour model imaginable. The method employs a recent differential renderer, without making any modifications to it, and simple MSE loss terms. The elegance of the model does not come at the expense of performance, and it achieves state of the art results on a wide variety of benchmarks, where in each benchmark, it outperforms the relevant deep learning baselines, as well as all classical methods.

## ACKNOWLEDGMENT

This project has received funding from the European Research Council (ERC) under the European Unions Horizon 2020 research and innovation programme (grant ERC CoG 725974). The contribution of the first author is part of a Ph.D. thesis research conducted at Tel Aviv University.

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
