# OpenReview forum: "End to End Trainable Active Contours via Differentiable Rendering"
_ICLR.cc/2020/Conference — Accept (Poster)_

### Official Review · AnonReviewer3 · 2019-10-20
**Official Blind Review #3**

**Rating:** 6

**Review:**

EDIT: The rating changed from '1: Reject' to '6: Weak accept' after the rebuttal. See below for my reasoning.

The submission considers two-class image segmentation problems, where a closed-contour image region is to be specified as the 'object'/region of interest, vs. 'no-object'/background. The approach taken here is end-to-end learning with an active-contour type approach. The main loss, in contrast to other active contour approaches, contains a direct difference of the estimated polygon area vs. ground truth polygon area.

The applied method seems conceptually quite simple (as admitted by the authors in Section 5), and the neural rendering approach seems quite neat, but both method presentation (Section 3) and evaluation (Section 4) seem incomplete and leave significant open questions.

One of my main concerns is related to the fact that the displacement field is static and, according to Figure 1 and Algorithm 1, is evaluated only once per image.
If the displacement field J is not conditioned on the current polygon shape (and this does not seem to be the case), then I am wondering why T iterations in the sampling/rendering part are necessary at all. When only considering L_seg, the optimal solution should be found within one iteration, since the displacement field will be able to provide the optimal answer. So maybe these iterations are only necessary when L_B and L_K are incorporated?
In any case, it is unclear why even L_seg is accumulated (using unweighted mean) over all T iterations before being backpropagated. Does this mean that these iterations are not meant to yield shape improvements? Why is ||M^t-M|| not evaluated per iteration, for the purpose of minimization?
It is also not sufficiently clear whether M^t in Equation 4 is a filled polygon mask, or if the mask is just related to the boundary (with a certain width). In absence of explanatory image material, I am assuming the former.
Overall the method description remains weak, since obvious questions/concerns such as the above are not addressed.

The experimental results look good from a quantitative point of view, and indeed, the strongest baselines, e.g. DARNet, are outperformed significantly in many cases.
Section 4 mostly focuses on quantitative evaluation and lots of picture examples, but fails to give insight into particular behaviors, failure cases, etc.
The evaluation procedure is cast a bit into doubt by two things: 1) In Figure 4, the initializations (blue circles) between the DARNet method and the proposed method are very different in size. I am wondering if this then still constitutes a fair comparison, and I have some doubts there. 2) In Figure 6, the proposed method consistently looks much worse than the DARNet baseline (and, in contrast to the baseline, completely fails for 4 vertices), unless the colors were swapped in the description.

Overall, I do not think the submission is in a good enough shape for acceptance.

Minor remarks:
- The values for lambda_1 and lambda_2 seem to come out of thin air, and they also seem quite small. It needs to be mentioned how they were determined.
- Data augmentation by rotation seems to be missing several values (between 270 and 260 degrees) and also not evenly spaced. Is this a typo or on purpose? In the latter case, an explanation is needed, since this seems weird.
- Section 4.3: There is no "Figure 4.2", I assume you mean Figure 6, which otherwise remains unreferenced.
- Section 4.3, Ablation Study: Don't use the word "derivatives" when you're talking about variations.
- Section 4.3, Ablation Study: "even without no auxiliary loss" -> remove "no" or change "without" -> "with"

-------------
Post-rebuttal comments:

I have read the revised version, as well as the other reviews and all authors' comments. The inclusion of an evaluation on a larger-size data set is highly appreciated, and seems to indeed validate the robustness of the method. Typos were fixed, including the switched color descriptions in Figure 7 (which should not have passed initial submission in the first place, if the text had been proofread properly).

Several of the open questions (e.g. "Why is L_seg accumulated before backpropagation?", "Why is the algorithm iterative if the displacement map is computed only once, if not for the other loss terms?", "Choice of values for lambda_1, lambda_2", Initial diameter of initialization") have been somewhat addressed by the authors in the rebuttal comment, though not in great detail.

Based on the quality of the results across data sets, and because I believe that the timely publication of this rather simple method can benefit further research in this area, I have adjusted my score to a 'Weak accept'. That said, I still do not think it is a good manuscript, and my score should be seen as a massive benefit of the doubt toward the authors.

Most importantly, above questions have NOT been adequately addressed in the actual revised text. The authors claim they have "improved the manuscript considerably", but yet I see more reasoning for certain choices described in the comment here than in the actual manuscript. Most of the changes are in Section 2 and the new Section 4.3, but not much relevant to my comments changed in Section 3.

For example, balloon and curvature losses aside, it is still not clear why an iterative approach would be helpful past the first iteration. An ideal displacement map that is not conditioned on the polygon should point, for each pixel, straight to the closest contour pixel. It is clear to me that this may not be what is being learned when multiple iterations are forced, yet it is not addressed why multiple iterations should be beneficial. (I could see why they could be beneficial if the approach was conditioned on the polygon vertices, to avoid vertex collapsing, but it's not.)

A good submission preempts these kinds of questions by addressing them carefully. What seems crystal clear to the authors will not be crystal clear to every reader. The authors should be more careful to include their reasoning in the actual text, which I believe this is essential for proper, easy understanding of the paper.

**Experience Assessment:**

I have published in this field for several years.

**Review Assessment: Checking Correctness Of Derivations And Theory:**

I carefully checked the derivations and theory.

**Review Assessment: Checking Correctness Of Experiments:**

I carefully checked the experiments.

**Review Assessment: Thoroughness In Paper Reading:**

I read the paper thoroughly.

---

> ### Author Response · Authors · 2019-11-08
> **We have improved the manuscript considerably following the feedback**
>
> Thank you for the detailed review.
>
> Indeed, our approach predicts the displacement field only once. We believe that this is a strength of our approach and is part of its simplicity. Similarly to RNNs with fixed weights, having the displacement map computed only once, does not mean that iterations are not beneficial. Note also that the error is backpropagated from all iterations.
>
> Regarding the use of the balloon and curvature term, please see the ablation study, which shows that while our method is extremely competitive even without these losses, the two losses contribute to the results.
>
> “Why is ||M^t-M|| not evaluated per iteration” -- As mentioned in the text before Eq.5., M^t is evaluated at each iteration given the updated set of points. Therefore, ||M^t-M|| is evaluated per iteration. The backpropagation is done on the accumulated loss.
>
> Clarity regarding M^t in Equation 4 - the mask M^t is a filled polygon rendered from the set of points P^t. We have further clarified this in the revision.
>
> We did not search for the best initial diameter, and simply fixed it to the size of 16 pixels across all datasets. Please note that DARNet uses multiple initializations (circles) or different sizes for each dataset as can be seen in Fig.4, while we use only one fixed-size circle. This further supports the robustness of our method.
>
> The caption of Fig.6 (of the original paper, 7 in the revised) is indeed a typo, and the colors were switched. We apologize for this and have fixed it in the revision. The quantitative results in the graphs of Fig.7 (of the original submission, now Fig. 8) support the fact that our method yields better segmentation for simple polygons as well.
>
> The values of Lambda1 and Lambda2 were fixed early during the development process and used across datasets. These reflect the relatively smaller part that the ballooning force and the curvature loss play, in the optimization. This is further supported by the ablation analysis that demonstrates that our method is extremely competitive even without these. Similarly, the set of rotations was set without much thinking early on during training, and since it worked, we kept it as is. We believe that changing the augmentation would contribute little to the results, and does not justify the pitfalls of multiple hypothesis testing.
>
> Overall, we hope that the simplicity and elegance of our method are not interpreted as a disadvantage. We believe that the power of our method over previous work (as complicated as they’ll be) is in the straightforward approach. Following the reviews, we have provided an additional dataset for comparison, and clarified and fixed the relevant sections and figures.
>
> With the CVPR deadline in a week, we would appreciate a timely response, in order for us to be able to plan our submission strategy.

---

> ### Author Response · Authors · 2019-11-11
> **Thank you for the additional feedback and for upgrading the paper’s rating**
>
> Thank you for the additional feedback and for upgrading the paper’s rating. We appreciate the timely response and apologize for neglecting to proofread the submitted version more carefully.
>
> Regarding the number of iterations. In the original (and revised) submission, Section 4.4 "Number of Iterations” and Fig.8 (as numbered in the revised version), we experiment with different number of iterations. We noticed that a single iteration is less beneficial across all datasets, while 2-3 iteration results in higher performance.  Nevertheless, as the reviewer hypothesised, a single iteration can already produce very good results, as can be seen from our experiments.
>
> We have released a new revision, elaborating on two subjects: (i) The initial guess, and (ii) the effect of the number of iterations T.

---

### Official Review · AnonReviewer2 · 2019-10-22
**Official Blind Review #2**

**Rating:** 8

**Review:**

This paper investigates an image segmentation technique that learns to evolve an active contour, constraining the segmentation prediction to be a polygon (with a predetermined number of vertices).  The advantage of active contour methods is that some shapes (such as buildings) can naturally be represented as closed polygons, and learning to predict this representation can improve over pixelwise segmentation.

The authors propose to learn an image-level displacement field to evolve the contour, and a neural mesh renderer to render the resulting mask for comparison with the ground truth mask.  The performance compared to prior learning-based active contour methods is impressive.

In section 4.3, there’s a reference to a “gap in performance” between the proposed method and DARNet and a reference to a "low number of vertices," but a comparison between the two methods as the numbers of vertices is varied seems to only be present in Fig. 6 -- it would be interesting to see an explanation of the discrepancy for the lower number of vertices seen in this figure.

Overall, due to the relative simplicity of the approach and impressive performance compared to prior learning-based approaches I recommend to accept.

Post-rebuttal:  I maintain my recommendation.

**Experience Assessment:**

I do not know much about this area.

**Review Assessment: Checking Correctness Of Derivations And Theory:**

N/A

**Review Assessment: Checking Correctness Of Experiments:**

I assessed the sensibility of the experiments.

**Review Assessment: Thoroughness In Paper Reading:**

I read the paper at least twice and used my best judgement in assessing the paper.

---

> ### Author Response · Authors · 2019-11-08
> **Thank you for the supportive review**
>
> Thank you for the supportive review.  We are sorry for the reference mistakes, these are all fixed in the new revised version.
>
> There was a typo in the caption of Fig.6 (Fig. 7 in the revised version), which switched the association between the methods and the colors. We believe that this mistake has led to the remark concerning this figure.

---

### Official Review · AnonReviewer1 · 2019-10-28
**Official Blind Review #1**

**Rating:** 8

**Review:**

The paper proposes a straightforward method for end-to-end learning of active contours, based on predicting a dense field of 2D offsets, and then iteratively evolving the contour based on these offsets. A differentiable rendering formulation by Kato et al is employed to make the process of aligning a contour to a GT mask differentiable.

The model shows rather compelling results on small datasets, and is very simple, with very strong parallels to active contours, which is a strength. The results improve those of DARNet, which to the best of my knowledge is the main published work in the space other than Curve-GCN. One thing that would be helpful, is  to have an experiment on a large dataset, such as Cityscapes -- right now all the datasets are testing the model in only the small-data regime. Perhaps in a supplement, it would also help to do ablation of how input image / dense deformation resolution affects the result quality -- the input can be subsampled by powers of 2 for the experiment.

As Amlan Kar helpfully points out, the work heavily overlaps with his approach "Fast Interactive Object Annotation with Curve-GCN", CVPR 2019, which is not cited or compared to. Curve-GCN similarly utilizes differential rendering (only a different variant) to match the GT masks. To me, the main difference wrt Curve-GCN is that explicit dense displacement fields are generated by the net and used directly for the iterative refinement steps, while Curve-GCN leverages implicit feature embeddings and uses GCN layers for their iterative updates. A second main difference is that Curve-GCN supports splines and interactive editing, while the proposed approach does not. Beyond these, there are multiple other differences that the authors point out, but those are more of a technical nature. Unfortunately, without a more direct comparison, it is very difficult to evaluate the design choices in the two approaches, which I feel is necessary for proper understanding of the paper.

AFTER REBUTTAL: The authors made additions that covered my concerns, so I have switched my recommendation.

A few more minor clarity / presentation issues.
-- “The recent learning-based approaches are either non-competitive or proven to be effective in the specific settings of building segmentation". It's not exactly clear what the point is in the context. Which "learning-based approaches"?
-- Typo 'backpropogation'.
-- A little better explanation of how a differentiable renderer of Kato works would have been helpful.
-- Figure 3 is not referenced in the text, takes a little bit of thought why it is relevant (helps explain Fig 1, but maybe better to show it prior to Fig 1).
-- In Eq 4 it’s not clear what F is.  (I see it is explained in Algorithm box, but that's much later)







**Experience Assessment:**

I have published in this field for several years.

**Review Assessment: Checking Correctness Of Derivations And Theory:**

I carefully checked the derivations and theory.

**Review Assessment: Checking Correctness Of Experiments:**

I carefully checked the experiments.

**Review Assessment: Thoroughness In Paper Reading:**

I read the paper at least twice and used my best judgement in assessing the paper.

---

> ### Author Response · Authors · 2019-11-08
> **We have improved the manuscript considerably following the detailed feedback**
>
> We thank the reviewer for the comprehensive review.
>
> We apologize for the typos in the previous draft. These have been corrected.
>
> To your comments:
>
> Comparison to Curv-GCN: as noted by the reviewer, the differences between the methods are in the support of splines and working with an embedding space in Curve-GCN, vs. displacement map. To emphasize: our method employs a single learned network that produces a displacement image in a single forward pass. The CNN used by Curve-GCN predicts an embedding space of size 28x28 that is further processed by graph neural networks.
>
> Following the review, we have conducted experiments on Cityscapes, which is the only public dataset available from Curve-GCN experiments (their code is not available). In this dataset, our method obtains SOTA for 6/8 classes and SOTA, by a sizable margin that is larger than the difference between the performance of previous work, in the overall mean mIoU. We believe that this also directly addressed the reviewer’s comment regarding larger datasets.
>
> All of our models are trained at the resolution of 64x64 pixels. As noted in the original submission when discussing the Vaihingen dataset “we experiment with different resizing factors during training”. Following the review, we share these results in Tab.5 of the revised submission. As can be seen, there is a steep degradation in performance below 64x64, which we attribute to the lack of details. When doubling the resolution to 128x128, the performance slightly degrades, however, that model could improve with further training epochs.
>
>
> The recent learning-based approaches are either non-competitive or proven to be effective in the specific settings of building segmentation" — we have clarified in the text that we mean learning-based active contour methods and have limited the scope of the claim.
>
> We have added a paragraph regarding the use of a 3D renderer for 2D maps. We simply fix the third coordinate and use the code of Kato as is.
>
> The letter “F” (for faces) is defined at the beginning of the “Method” section.
>
> We believe that various issues raised by the reviewer were fully addressed in a way that considerably improved the manuscript. With the CVPR deadline in a week, we would appreciate a timely response, in order for us to be able to plan our submission strategy.

---

### Public Comment · ~Amlan_Kar2 · 2019-10-22
**Related Work**

Thanks for the nice work! I would like to point out our related work [1], published at CVPR '19 here, that also utilizes a differentiable renderer to render polygons into masks similar to your work. It would be nice to discuss contributions in light of our paper as well, thanks!

[1] Fast Interactive Object Annotation with Curve-GCN: https://arxiv.org/abs/1903.06874 - CVPR 2019

---

> ### Author Response · Authors · 2019-10-24
> **Thank you for pointing us to the missing related work**
>
> Thank you very much for pointing us to [1], which is indeed related and would be cited appropriately. We would like to enumerate some of the important differences between the methods.
> 1. Supervision and training: We supervise with GT masks only during training, while [1] learns an additional edge branch and a vertex branch. Vertex-based supervision constraints their model to learn a specific location for each point.
> 2. Training: While we train our model end-to-end with a single supervision, [1] performs a two-phase learning, where they first train using edges and vertices supervision, followed by fine-tuning with GT masks. [1] also points out that their rendering process is too slow for training end-to-end using only GT masks (Sec. “Training Details”), while we use a fast, fully differentiable renderer.
> 3. CNN role: Our method employs a single learned network that produces a displacement image in a single forward pass. The CNN used by [1] predicts an embedding space of size 28x28 that is further processed by other networks.
> 4. CNN architecture: We employ a fully convolutional CNN that produces an output that is the same size as the input image, while [1] scales a fixed-sized input to a spatially-limited 28x28 image.
> 5. To emphasize: our method is considerably more direct, and learns a 2-D displacement field in the scale of the input by a fully convolutional network. The update in [1] is by a learned GCN that is applied over graph nodes that employ the 28x28 embedding.
> 6. Loss: we incorporate two loss terms that are based on time-tested pulling forces from the classical active contour literature: the Balloon and Curvature terms. This allows us to work directly with the contour.
>
> [1] Fast Interactive Object Annotation with Curve-GCN: https://arxiv.org/abs/1903.06874 - CVPR 2019

---

### Author Response · Authors · 2019-11-08
**A revised version, including new experiments comparing with Curve-GCN**

Following the reviews, we have revised our manuscript to correct the various typos, to clarify some issues and, most importantly, to update the related work section and to compare experimentally with the CVPR 2019 work of Ling et al. As detailed on open review, which this work indeed narrows our novelty claims, there are important differences and the two methods are very much different.

We are happy to report that on the public dataset on which the CVPR 2019 work has been tested, our method outperforms all previous work in 6/8 categories and shows a clear advantage in the mean performance. This, without performing any modification to our method and despite our method being considerably less involved than the other methods.

---

### Public Comment · ~Ali_Hatamizadeh1 · 2019-12-22
**More Related Work**

We appreciate your work and would like to bring to your attention our paper published as:

@article{hatamizadeh2019endtoend,
title={End-to-End Deep Convolutional Active Contours for Image Segmentation},
author={Ali Hatamizadeh and Debleena Sengupta and Demetri Terzopoulos},
journal={arXiv preprint arXiv:1909.13359},
month={September},
year={2019}
}

It introduced the first fully automatic, end-to-end trainable CNN-ACM combination, where the Active Contour Model (ACM) is defined implicitly, as a level set. This has important advantages relative to your use of the explicit ACM formulation. Among them is the fact that our DCAC model can simultaneously segment multiple object instances, as opposed to just a single instance, while dealing with arbitrary shapes and capturing sharp corners as necessary. Our DCAC model is implemented entirely in Tensorflow and is thus end-to-end differentiable and backpropagation trainable. It requires no user intervention either during training or during image segmentation. We trained and tested DCAC on the Vaihingen and Bing Huts datasets and our results established a new state-of-the-art performance by a wide margin at the time (March 2019).

Our foregoing publication is highly relevant to your work and should be discussed in your Related Work section, such as in your paragraph on "Building segmentation and recent active contour solutions ". Thanks.

---

> ### Author Response · Authors · 2020-01-05
> **Post-submission related-work**
>
> Thank you for letting us know about your paper, which we will cite in the next version.
>
> We ask that you would also cite our work, noting that it was made public on open review before the publication date of your arxiv manuscript (obviously the two efforts are concurrent).

---

> > ### Public Comment · ~Ali_Hatamizadeh1 · 2020-01-12
> > **Related Work**
> >
> > Thank you. We originally submitted our paper, as it appears on arXiv, to ICCV 2019, back in March 2019, but it was ultimately rejected for perplexing reasons, despite the fact that our model significantly outperformed the then state-of-the-art building segmentation methods. We will cite your paper in the published version of our work.

---

> > > ### Author Response · Authors · 2020-01-14
> > > **Related work**
> > >
> > > This is unfortunate and does not seem right. We will treat your paper as concurrent to our work and think that it should be treated as such by all future reviewers.

---

### Public Comment · ~Gladis_Ne_Limes1 · 2023-07-17
**re**

Schedule interviews with shortlisted candidates to evaluate their technical skills, problem-solving abilities, and communication. 9. Check References: If possible, reach out to the candidate's references to get insights into their work ethics and performance. 10. NDA and Contract: If you decide to move forward with a candidate, ensure you have a well-defined contract that includes terms, project milestones, payment details, and any non-disclosure agreements (NDAs) if needed. 11. Collaboration Tools: Set up communication and collaboration tools to facilitate smooth project management and communication with the hired Android app developer https://mlsdev.com/blog/hire-android-developer. Remember that hiring the right Android app developer is crucial for the success of your project. Take your time, ask relevant questions, and ensure you find a candidate or team with the necessary skills and experience to deliver a high-quality Android application.

---

### Decision · Program_Chairs · 2019-12-19

**Decision:**

Accept (Poster)

**Comment:**

The submission presents a differentiable take on classic active contour methods, which used to be popular in computer vision. The method is sensible and the results are strong. After the revision, all reviewers recommend accepting the paper.